# Unsupervised Learning of Disentangled Representation via Auto-Encoding: A Survey

**DOI:** 10.3390/s23042362

**Published:** 2023-02-20

**Authors:** Ikram Eddahmani, Chi-Hieu Pham, Thibault Napoléon, Isabelle Badoc, Jean-Rassaire Fouefack, Marwa El-Bouz

**Affiliations:** 1L@bISEN, LSL Team, Yncrea Ouest, 29200 Brest, France; 2Generix Group, 75012 Paris, France; 3LaTIM, INSERM UMR1101, University of Brest, 29200 Brest, France; 4L@bISEN, VISION-AD Team, Yncrea Ouest, 29200 Brest, France

**Keywords:** representation learning, disentanglement, auto-encoder, generative models, neural networks, metrics

## Abstract

In recent years, the rapid development of deep learning approaches has paved the way to explore the underlying factors that explain the data. In particular, several methods have been proposed to learn to identify and disentangle these underlying explanatory factors in order to improve the learning process and model generalization. However, extracting this representation with little or no supervision remains a key challenge in machine learning. In this paper, we provide a theoretical outlook on recent advances in the field of unsupervised representation learning with a focus on auto-encoding-based approaches and on the most well-known supervised disentanglement metrics. We cover the current state-of-the-art methods for learning disentangled representation in an unsupervised manner while pointing out the connection between each method and its added value on disentanglement. Further, we discuss how to quantify disentanglement and present an in-depth analysis of associated metrics. We conclude by carrying out a comparative evaluation of these metrics according to three criteria, (i) modularity, (ii) compactness and (iii) informativeness. Finally, we show that only the Mutual Information Gap score (MIG) meets all three criteria.

## 1. Introduction

Data representation is a crucial and long-standing issue in machine learning, as it has a significant impact on model performance [1]. For that reason, much of the actual efforts in the machine learning community have been toward representation learning [2,3,4,5].

Representation learning refers to finding a low-dimensional representation that captures true underlying factors of variation that explain the data [6]. A series of traditional statistical approaches have been reported for the estimation of such a low-dimensional representation, such as Principal Component Analysis (PCA) [7,8], Independent Component Analysis (ICA) [9,10] or Single Value Decomposition (SVD) [11]. They aim to identify the underlying factors of variation in the data. However, in practice, the data to be encoded may be very large in dimension and contain factors that cannot be captured with these linear methods.

Disentangled representation learning has emerged as an effective way of finding a low-dimensional space of complex data while addressing the problem of identifying the independent factors of variation. Following the definition of Bengio et al. [3]: “a disentangled representation is a representation where a change in one latent variable corresponds to a change in one generative factor, while being relatively invariant to changes in other factors”. As an example, a model trained on a set of face images can capture different generative factors, i.e., pose, gender, skin color or smile, and encode them into independent latent variables in the representation space. Each latent variable is sensitive to a change in only one generative factor. Let $z_{k}$ be the latent factor controlling the facial pose, then varying $z_{k}$ while fixing other factors would generate images of different facial poses but with the same other generative factors (gender, skin, color, smile). The same goes for latent variables controlling other generative factors. We illustrate the notation of generative factors and latent factors in Figure 1.

These representations have been useful for model generalization by discovering the causal variables in data and capturing its compositional structure [12,13,14,15], allowing the learning systems to understand real-world observations as humans do [16], and, therefore, representation learning could generalize to unseen scenarios. For example, a model trained to generate an image of a green square and blue triangle, because of the generalizability property, the model can also generate a blue square and green triangle [17]. Following this motivation, they have been of interest to downstream tasks, such as supervised learning, compression and data augmentation. In supervised learning, it can be used as an input feature when building classifiers or other predictors to improve predictive performance [18], reduce sample complexity [19] and offer interpretability [20]. For compression [21], disentangled representations are compact and low-dimensional, thus minimizing the cost associated with storing underlying factors of variation in data. Further, they can be used to generate novel examples not found in the original dataset [21]. Such feature learning also supports a variety of other applications, such as super-resolution [22], multimodal application [23,24,25,26,27], medical imaging [28,29], video prediction [30,31,32,33,34], natural language processing [35,36,37], transfer learning and zero-shot learning [38].

A large range of state-of-the-art methods for learning unsupervised representations is based on variational auto-encoders (VAE) [39]. Variational auto-encoders have been shown to be useful for learning high-dimensional data and inferring latent variables. However, they often fail to capture a disentangled representation of the data [20]. In order to overcome these drawbacks, several variants of VAE have been proposed [40,41,42,43,44] with the idea that they could allow better disentanglement [45,46]. Another line of work in this field is based on Generative Adversarial Networks (GAN) [47,48]. Numerous variants of GAN have been proposed and demonstrated the ability to learn a disentangled representation [49,50,51,52,53] and were reported to have comparable performance to VAE-based methods [53]. A relative field with disentangled representation learning is Self-Supervised Learning (SSL) [54,55,56]. Self-supervised learning provides a way for learning representation from unlabeled data. Recent efforts have been made toward using self-supervised algorithms in order to learn a disentangle representation [57,58,59,60]. However, recent studies have reported that the existing SSL methods often struggle to learn disentangled representations of the data [60].

In this paper, we aim to provide a systematic and comprehensive survey of VAE-based approaches. We attempt to shed light on some VAE variants that are considered state-of-the-art of disentangled representations and to provide an analysis of the common idea behind all these approaches. Further, we conduct an extensive review of disentanglement metrics, where we explore what makes one metric better than another. Finally, we discuss limitations and future directions in this field of research. We sum up the recent work focusing on disentanglement and the metric proposed alongside each method in Table 1.

To the authors’ best knowledge, [61] is the only study focusing on disentanglement representation methods from a practical point of view. In [61], the authors introduce a library to train and evaluate disentangled representations. However, a detailed description of the methods is missing. On the other hand, no other study performs this type of review focused on grouping disentanglement methods as well as metrics and presents a theoretical analysis and a detailed description of each method and their added value. The purpose of this survey is to provide researchers interested in this broad field with a comprehensive overview of state-of-the-art approaches and establish a guideline to choose a suitable approach given an objective.

The rest of this paper is organized into six sections. In Section 2, we describe the main concepts that are necessary to understand the methods considered in this work. In Section 3, we review unsupervised disentanglement methods based on the auto-encoder baseline. In Section 4, we review the most well-known metrics to evaluate disentanglement. In Section 5, we present a detailed discussion of disentanglement methods and a comparison between metrics. Finally, we conclude the work and discuss its future scope in Section 6.

## 2. Background

In this section, we provide a detailed description of several notions that will be found throughout this paper.

### 2.1. Auto-Encoders

An auto-encoder [1,3,62,63,64] is a neural network architecture that is trained to reconstruct its input [65] with the least possible amount of distortion. Their main purpose is to learn a compressed meaningful representation of the data that can be used for various applications, including clustering [66] and classification [67,68].

Here we briefly describe the auto-encoder (AE) framework:

Encoder: A neural network *f* that maps an input (image, tensor, curve) into a hidden representation *Z* capturing the significant underlying factors of the data, also called an inference model. Given a data set, $\left\{x_{1},\ldots ,x_{T}\right\}$, for each $x_{i}$, we define: (1)$$z_{i}=f\left(x_{i}\right),$$

Latent space: A low-dimensional representation of the data. The vector *z* is the feature-vector, also called latent code or latent dimension. *z* is called “latent” because it is a variable produced by the model from the input data.

Decoder: A neural network *g* that builds back the input from its latent vectors.
(2)$$\tilde{x_{i}}=g\left(z_{i}\right),$$

Training an auto-encoder consists of learning the functions *f* and *g* that minimize the error E of the reconstruction loss function △, which measures the difference between the input and its reconstruction: (3)$$\arg\min_{f,g}E\left[△\left(x_{i},g\left(f\left(x_{i}\right)\right)\right)\right]$$

### 2.2. Variational Auto-Encoders (VAEs)

Kingma et al. [39] introduce a stochastic variational inference for an auto-encoder. Variational auto-encoders attempt to describe data generation through a probabilistic modeling perspective [69]. In VAE, inputs are encoded as a distribution over latent space instead of as single points [70]. In doing so, Kingma et al. assume a posterior distribution on the latent variable $z_{i}$ for each data point $x_{i}$ denoted by inference model $q_{\phi }\left(z|x\right)$. This inference model corresponds to the probabilistic encoder, and is parameterized by ϕ [65]. In a similar vein, they introduce a generative model $p_{\theta }\left(x|z\right)$, which is equivalent to a probabilistic decoder determined by the parameter θ: given a latent variable *z* it returns a distribution on the corresponding possible values *x*. Finally, they consider a prior distribution over the latent variables $z_{i}$ denoted by $p_{\theta }\left(z_{i}\right)$, where $p_{\theta }\left(z\right)$ is a standard multivariate normal distribution N(0,I) and *I* is the identity matrix. Training a VAE consists of simultaneously learning the parameters ϕ and θ. One way to estimate these parameters is to use the maximum log-likelihood (ML), a common criterion for probabilistic models.

The marginal log-likelihood is the sum of each data points $\log p_{\theta }\left(x_{1},\ldots ,x_{N}\right)=\sum_{i=1}^{N}\log p_{\theta }\left(x_{i}\right)$. Each point can be rewritten as [65]:(4)$$\log p_{\theta }\left(x_{i}\right)=D_{KL}\left(q_{\phi }\left(z|x_{i}\right)\left|\right|p_{\theta }\left(z|x_{i}\right)\right)+L\left(\theta ,\phi ,x_{i}\right)$$

The first term in Equation (Equation 4) is the Kullback–Leibler (KL) divergence, which determines the distance between the approximate posterior and the true posterior, and the second term is the variational lower bound defined in [69] as: (5)$$L\left(\theta ,\phi ,x_{i}\right)=E_{q_{\phi }\left(z|x_{i}\right)}\left[-\log q_{\phi }\left(z|x\right)+\log p_{\theta }\left(x,z\right)\right]$$

Since the KL divergence is non-negative, $L(\theta ,\phi ,x_{i})$ is the lower marginal log-likelihood bound, also known as the Evidence Lower Bound Objective (ELBO). This can be further expressed as: (6)$$L_{VAE}≃L\left(\theta ,\phi ,x_{i}\right)=E_{q_{\phi }\left(z|xi\right)}\left[\log p_{\theta }\left(x_{i}|z\right)\right]-D_{\mathrm{KL}}\left(q_{\phi }\left(z|x_{i}\right)\left|\right|p_{\theta }\left(z\right)\right)$$

Relying on Equation (Equation 6), one can notice that the evidence lower bound is a sum of two terms: the first term is a negative reconstruction error that needs to be maximized in order to increase the reconstruction capability of the sample, and the second term is the KL divergence that acts as a regularizer to ensure that the approximate posterior $q_{\phi }\left(z|x\right)$ remains close to the prior. Finding the model parameters θ and ϕ that will maximize the marginal likelihood of the data while simultaneously minimizing the KL divergence between the approximation $q_{\phi }\left(z|x\right)$ and the prior $p_{\theta }\left(z\right)$ is equivalent to maximizing the ELBO.

It can thus be said that training a variational auto-encoder consists of maximizing the variational lower bound objective. The ELBO (Equation (Equation 6)) serves as the core of the variational auto-encoder and the methods we will discuss in the rest of this paper, so it is worth spending some thinking about how it can be optimized [71]: (7)$$\max_{\phi ,\theta }E_{p(x_{i})}\left[E_{q_{\phi }\left(z|x_{i}\right)}\left[\log p_{\theta }\left(x_{i}|z\right)\right]-D_{KL}(q_{\phi }\left(z|x_{i}\right)\left|\right|p_{\theta }\left(z\right))\right]$$

As is common in machine learning, the ELBO can be optimized with regard to all parameters (ϕ and θ) using stochastic gradient descent [72], but first, more detail about $q_{\phi }\left(z|x_{i}\right)$ is required. The usual choice is a simple factorized Gaussian encoder $q_{\phi }\left(z|x_{i}\right)∼N\left(\mu_{\phi }\left(x_{i}\right),\sigma_{\phi }\left(x_{i}\right)\right)$, where $\mu_{\phi }\left(x_{i}\right)$ and $\sigma_{\phi }\left(x_{i}\right)$ are the mean and standard deviation implemented via neural networks and $\sigma_{\phi }\left(x_{i}\right)$ is constrained to be a diagonal matrix. Under this choice, we have a KL divergence between two Gaussian distributions, which is tractable. Hence we can calculate the gradient of the last term of Equation (Equation 7). However, the first term is a bit trickier as it requires an estimation by sampling from $q_{\phi }\left(z|x_{i}\right)$. Kingma et al. [39,72] propose to estimate the marginal likelihood lower bound of the full dataset using mini-batches of *M* data-points and then average the gradient over these mini-batches. However, stochastic gradient descent via back-propagation cannot handle stochastic variables within the network. To solve this problem and generalize back-propagation through random sampling, they propose another way to generate samples from $q_{\phi }\left(z|x_{i}\right)$; this solution is called the “reparameterization trick” [39,73]. The key behind this trick is to define *z* as a deterministic function $z=g(\phi ,x_{i,\xi })$, then sample from the posterior $q_{\phi }\left(z|x_{i}\right)$ using $z=\mu \left(x_{i}\right)+\sigma \left(x_{i}\right)*\xi$. Here, ξ is an auxiliary variable with independent marginal $p\left(\xi \right)∼N\left(0,I\right)$, and $g_{\phi }\left(.\right)$ is a vector-valued function parameterized by ϕ. In doing so, we are keeping the stochasticity of the variables, but also, we have a deterministic function of inputs that will work for stochastic gradient descent. The architecture of a variational auto-encoder (VAE) is shown in Figure 2.

For ease of reference, we sum up in Table 2 the main terms and corresponding mathematical symbols used in this work.

### 2.3. Reconstruction Error

In order to make the optimization of the evidence lower bound objective (Equation (Equation 7)), the posterior $q_{\phi }\left(z|x_{i}\right)$ is pushed to match the unit Gaussian prior $p_{\theta }\left(z\right)∼N\left(0,I\right)$. Since the posterior $q_{\phi }\left(z|x_{i}\right)$ and the prior $p_{\theta }\left(z\right)$ are factorized (i.e., have diagonal covariance matrix) and the samples from $q_{\phi }\left(z|x_{i}\right)$ are generated using the reparameterization trick, learning a representation of the data depending only on $q_{\phi }\left(z|x_{i}\right)$ may result in a meaningless representation where only a limited number of latent variables are exploited for data reconstruction. In doing so, the amount of information that can be transmitted through the latent channels is reduced. Thus, this results in high reconstruction errors and low reconstruction fidelity [74].

### 2.4. Mutual Information Theory

Mutual information is a fundamental quantity for measuring dependency between random variables [75]. Let (*X*, *Z*) be a couple of random variables. The mutual information (MI) between *X* and *Z*, denoted as I(X;Z), is: (8)$$I\left(X;Z\right)=D_{KL}\left(P_{XZ}\left|\right|P_{X}⊗P_{Z}\right)$$
where $D_{KL}$ is the Kullback–Leibler (KL) divergence between the joint distribution and the product of the marginals.

## 3. Methods

The major challenge behind representation learning is how we can choose the model that leads to better disentanglement and thus can be useful for later downstream tasks. In this section, we present an overview of the state-of-the-art frameworks in representation learning based on auto-encoding (see Table 1).

### 3.1. β-Variational Auto-Encoder

Higgins et al. [20] introduce a variant of variational auto-encoders [39], β-VAE, a deep generative (unsupervised) algorithm for learning disentangled representations. The authors propose to modify the evidence lower bound objective (ELBO) by up weighting the KL divergence term in Equation (Equation 7) in order to learn a disentangled representation: (9)$$\max_{\phi ,\theta }E_{p(x_{i})}\left[E_{q_{\phi }\left(z|x_{i}\right)}\left[\log p_{\theta }\left(x_{i}|z\right)\right]-\beta D_{KL}\left(q_{\phi }\left(z|x_{i}\right)\left|\right|p_{\theta }\left(z\right)\right)\right]$$
where β is an adjustable hyper-parameter higher than 1. It can be noted that β-VAE with β=1 is equivalent to the original VAE framework [39]. Re-writing the ELBO for β-VAE: (10)$$L_{\beta \text{-}\mathrm{VAE}}=E_{q_{\phi }\left(z|x_{i}\right)}\left[\log p_{\theta }\left(x_{i}|z\right)]\;-\;\beta D_{KL}\left(q_{\phi }\left(z|x_{i}\right)\left|\right|p_{\theta }\left(z\right)\right)\right]$$

Such a penalization causes the posterior $q_{\phi }\left(z|x\right)$ to better match the factorized prior $p_{\theta }\left(z\right)$, which is associated with the need to maximize the log-likelihood of data *x* and push the model to learn a disentangled representation of the data.

β-VAE was performed using a number of benchmarks with known ground truth factors, such as CelebA [76] (202,599 color images of celebrity faces), Chairs [77] (86,366 color images of chairs), Faces [78] (239,840 gray-scale images of 3D faces) and 2D Shape [79] (737,280 synthetic images of 2D shapes such as heart, oval and square). The size of the images in the four datasets is 64×64 pixels.

### 3.2. InfoMax-Variational Auto-Encoder

As we can see from Equation (Equation 10), learning representation depending only on $q_{\phi }\left(z|x\right)$ may result in meaningless representations and, therefore, a poor reconstruction quality. To avoid collapsed representations, Rezaabad et al. [44] report a simple yet practical method to build a meaningful representation. They propose to extend the evidence lower bound (ELBO) with a regularizer term that maximizes the mutual information between the data and the latent representation. In so doing, the model is pushed to maximize the information about the data (input) stored in the inferred representation (latent representation) [44]. This solution is referred to as InfoMax-VAE. They end up with the following ELBO: (11)$$\max_{\phi ,\theta }E_{q(x)}\left[L_{\beta \text{-}\mathrm{VAE}}\right]+\alpha I_{q_{\phi }}\left(x,z\right)$$
where β and α≥0 are regularization coefficients for the KL divergence and mutual information. Varying α controls the amount of information stored in the latent representation, also known as information preference.

Following [80], the mutual information is estimated by the average KL divergence between the joint and associated marginals: $(I_{q_{\phi }}\left(x,z\right)=KL\left(q_{\phi }\left(x;z\right)\left|\right|q\left(x\right)⊗q_{\phi }\left(z\right)\right)$. However, the KL divergence is hard to compute in general due to the intractable posterior, so Rezaabad et al. argue [44] that since KL divergence comes from a large class of different divergence, we can replace this term with another variational *f*-divergence $D_{f(t)}$ where *t* represents all possible functions. Thus, they arrive at the following equation: (12)$$\max_{\phi ,\theta }E_{q(x)}\left[L_{\beta \text{-}\mathrm{VAE}}\right]+\alpha D_{f}\left(q_{\phi }\left(x,z\right)\left|\right|q\left(x\right)q_{\phi }\left(z\right)\right)$$

Specifically, they choose f(t) to be tlogt (more details about this choice can be found in [44]), arriving to the final ELBO for InfoMax-VAE: (13)$$L_{\mathrm{InfoMax}\text{-}\mathrm{VAE}}=L_{\beta \text{-}\mathrm{VAE}}+\alpha \left(E_{q_{\phi }\left(x,z\right)}\left[t(x,z)\right]-E_{q\left(x\right)q_{\phi }\left(z\right)}\left[\exp\left(t(x,z)-1\right)\right]\right)$$

This leaves the task of evaluating $E_{q_{\phi }\left(x,z\right)}$ and $E_{q(x)q(z)}$. They propose a simple yet practical way to do so: first of all, and thanks to the reparameterization trick [72], they draw samples from q(x), ($x_{i},z_{i})∼q_{\phi }\left(x,z\right)=q_{\phi }\left(z|x\right)q\left(x\right)$. Afterward, to get samples from the marginal $q_{\phi }\left(z\right)$, they choose a random data point $x_{j}$, followed by sampling from $z∼q_{\phi }\left(z|x_{j}\right)$.

The InfoMax-VAE was performed using the CelebA dataset and has been shown to be capable of learning meaningful and disentangled representation and outperforms β-VAE.

### 3.3. Factor Variational Auto-Encoder

Kim and Mnih [42] adopt another decomposition of the ELBO, specifically, they decompose the KL term in Equation (Equation 6) as Hoffman and Johnson propose in [81,82]: (14)$$E_{p_{data}\left(x\right)}[D_{KL}\left(q\left(z|x\right)||p\left(z\right))\right]=I\left(x;z\right)+D_{KL}\left(q\left(z\right)||p\left(z\right)\right)$$
where I(x;z) is the mutual information between *x* and *z* and $q\left(z\right)=E_{p_{data(x)}}\left[q\left(z|x\right)\right]=\frac{1}{N}\sum_{i=1}^{N}q\left(z|x_{i}\right)$ is the latent distribution for all data. Penalizing $D_{KL}\left[q\left(z\right)||p\left(z\right)\right]$ encourages q(z) to match the factorized prior p(z) and, therefore, encourages disentanglement. Further, penalizing I(x;z) reduces the information about the data *x* kept in the latent space *z*, which might result in less accurate reconstructions for high values of β.

Based on this observation, Kim and Mnih [42] argue that it may not be necessary or desirable to penalize the mutual information between *x* and *z* in order to have a better disentanglement. However, they propose to add an additional term to the VAE objective (Equation (Equation 6)) that penalizes the dependence of variables within the latent space [46]: (15)$$E_{p(x)}[E_{q_{\phi }\left(z|x\right)}\left[\log p_{\theta }\left(x|z\right)\right]-D_{KL}(q_{\phi }\left(z|x\right)||p\left(z\right))]-\gamma D_{KL}\left(q\left(z\right)\right)\left|\right|\prod_{i=1}^{d}q\left(z_{j}\right)$$

Re-writing the ELBO for Factor-VAE: (16)$$L_{\mathrm{Factor}\text{-}\mathrm{VAE}}=L_{\mathrm{VAE}}-\gamma D_{KL}\left(q\left(z\right)|\right|\prod_{i=1}^{d}q\left(z_{j}\right))$$

The second term is total correlation (TC) [83], a general measure of dependence between several random variables. This term is intractable since the estimation of both q(z) and $q(z_{j})$ requires a pass through the entire data set. Hence Kim and Mnih [42] propose another alternative for optimizing this term using the density ratio trick [84,85]. The density ratio trick consists of training a binary classifier/discriminator that returns the probability d(z) that its input was sampled from q(z) rather than from $q(z_{j})$: (17)$$TC\left(z\right)=D_{KL}(q\left(z\right)||q\left(z_{j}\right))\;=E_{q(z)]}\left[\log\frac{q(z)}{q(z_{j})}\right]≃E_{q(z)}\left[\log\frac{d(z)}{1-d(z)}\right]$$

The VAE and the discriminator are trained jointly. The VAE parameters are fine-tuned using the objective in Equation (Equation 16) with the total correlation term changed to its approximation in Equation (Equation 17).

Factor-VAE was performed using several benchmarks, such as CelebA, Chairs and Faces. Moreover, two synthetic datasets were used: 2D Shapes and 3D Shapes containing 480,000 64×64×3 RGB images of 3D shapes [79].

### 3.4. β-Total Correlation Variational Auto-Encoder

Concurrently to Kim and Mnih [42], Chen et al. [41] proposed another approach that surpasses both β-VAE performance and Factor-VAE complexity. β-TCVAE is a deep unsupervised approach for learning disentangled representation, a replacement of β-VAE, with no additional hyper-parameters during training.

Chen et al. [41] proposed a different decomposition of the second term in Equation (Equation 10), arriving at the following split up: (18)$$\begin{array}{cc} E_{p(x)}[D_{KL}(q_{\phi }\left(z|x\right)||p\left(z\right))]= & D_{KL}\left(q\left(z,x\right)||q\left(z\right)p\left(x\right)\right)+D_{KL}\left(q\left(z\right)|\right|\prod_{j}q\left(z_{j}\right))\\ \; & +\sum_{j}D_{KL}(q\left(z_{j}\right)||p\left(z_{j}\right)) \end{array}$$

The first term is known as the index-code mutual information (MI), which denotes the mutual information between the data and the latent space. The second term is total correlation (TC). The last term is the dimension-wise KL, which primarily encourages the latent dimensions to better match their corresponding priors.

According to Chen et al. [41], the total correlation term in the ELBO is the one that affects disentanglement. To verify this claim, the TC-term is evaluated using a Monte-Carlo approximation [86].

A unique integer index is assigned to each training sample, and they use the following estimator given a mini-batch of samples $\left\{n_{1},n_{2},\ldots ,n_{M}\right\}$: (19)$$E_{q(z)}\left[\log q(z)\right]≃\frac{1}{M}\sum_{i=1}^{M}\left[\log\frac{1}{NM}\sum_{j=1}^{M}q\left(z\left(n_{i}\right)|n_{j}\right)\right]$$
where $q(z|n_{i})$ is close to 0 for a randomly sampled component but large if *z* comes from component $n_{i}$.

To achieve better disentanglement, the authors up-weight each term of the ELBO individually. Re-writing the β-TCVAE objective: (20)$$L_{\beta \text{-}\mathrm{TCVAE}}=L_{\beta \text{-}\mathrm{VAE}}-\alpha I_{q}\left(z;n\right)-\beta KL(q\left(z\right)||\prod_{j}q\left(z_{j}\right))$$

β-TCVAE was performed using the same datasets as Factor-VAE (CelebA, Chairs, Faces and 3D Shapes).

### 3.5. DIP-Variational Auto-Encoder

Another line of work has argued that pushing the posterior $q_{\phi }\left(z\right)$ to match a factorized prior p(z) can lead to a better disentanglement. Kumar et al. [43] added a regularizer to the ELBO to encourage disentanglement during inference, therefore: (21)$$L_{\mathrm{VAE}}-\lambda D\left(q\left(z\right)||p\left(z\right)\right)$$
where λ is a hyper-parameter controlling its effect on the evidence lower bound objective, and *D* is an (arbitrary) divergence. In order to estimate this term, they suggest matching the moments of these distributions. In particular, they propose to penalize the $\ell_{2}$ distance between $q_{\phi }\left(z\right)$ and N(0,1) in order to match their covariances.

Let us denote: (22)$$Cov_{q_{\phi }\left(z\right)}\left[z\right]=E_{p(x)}Cov_{q_{\phi }\left(z|x\right)}\left[z\right]+Cov_{p(x)}\left(E_{q_{\phi }\left(z|x\right)}\left[z\right]\right)$$
where $E_{q_{\phi }\left(z|x\right)}\left[z\right]$ and $Cov_{q_{\phi }\left(z|x\right)}\left[z\right]$ are random variables that are functions of random variable *x*. Since $q_{\phi }\left(z|x\right)∼N\left(\mu_{\phi }\left(x\right),\sum \sigma \left(x\right)\right)$, Equation (Equation 22) becomes: (23)$$Cov_{q_{\phi }\left(z\right)}\left[z\right]=E_{p(x)}\left[\sum \sigma_{\phi }\left(x\right)\right]+Cov_{p(x)}\left[\mu_{\phi }\left(x\right)\right]$$

Kumar et al. [43] explored two options for disentangling regularizers to get this term close to the identity matrix: (i) regularizing the deviation of $Cov_{p(x)}\left[\mu_{\phi }\left(x\right)\right]$ from the identity matrix, which they refer to as DIP-VAE-*I*. (ii) regularizing $Cov_{q_{\phi }\left(x\right)}\left[z\right]$, which they denote as DIP-VAE-II.

Maximizing the objective of either DIP-VAE-*I* Equation (Equation 24) or DIP-VAE-II Equation (Equation 25) leads to better disentanglement.
(24)$$L_{\mathrm{DIP}\text{-}\mathrm{VAE}\text{-}I}=L_{\mathrm{VAE}}-\lambda_{1}\sum_{i\neq j}{\left[Cov_{p(x)}{\left[\mu_{\phi (x)}\right]}_{}\right]}_{ij}^{2}-\lambda_{2}\sum_{i}{\left({\left[Cov_{p(x)}\left[\mu_{\phi (x)}\right]\right]}_{ii}-1\right)}^{2}$$
(25)$$L_{\mathrm{DIP}\text{-}\mathrm{VAE}\text{-}\mathrm{II}}=L_{\mathrm{VAE}}-\lambda_{1}\sum_{i\neq j}{\left[Cov_{q_{\phi }}{\left[z\right]}_{ij}\right]}^{2}-\lambda_{2}\sum_{i}{\left({\left[Cov_{q_{\phi }}\left[z\right]\right]}_{ii}-1\right)}^{2}$$

DIP-VAE was performed using three datasets: CelebA, 3D Chairs and 2D Shapes. DIP-VAE has been shown to be superior to β-VAE and capable of learning disentangled factors without having any conflict between disentanglement and quality reconstruction.

To fairly evaluate such methods, the authors have chosen the same CNN architecture. Table 3 describes the experimental details, including the encoder and decoder architectures:

## 4. Metrics

In order to evaluate the approaches described above and use them for downstream tasks, a metric of disentanglement is required. Most prior works relied on a visual inspection of the latent representation [87], but recently more rigorous metrics have been proposed.

To the authors’ best knowledge, [88,89] are the only studies analyzing disentanglement metrics. In this review, we choose to focus on discussing the metrics proposed alongside each method above Table 1), clarifying their strengths and shortcomings.

### 4.1. $Z_{diff}$ Score

Higgins [20] introduced a disentanglement metric called *Z*-diff, also known as the β-VAE metric based on the following intuition: if one generative factor is fixed while randomly sampling all others, we will have a disentangled representation in which the latent variable corresponding to the fixed generative factor will vary less than the others. Applying this metric involves following these steps:Randomly select a generative factor $f_{k}$.Create a batch of couples vectors, $p_{1}$ and $p_{2}$, where the value of the chosen factor $f_{k}$ is kept fixed and equal within the pair while the other generative factors $f_{k-1}$ are chosen randomly. For a batch of *L* samples:
$$p_{1}=\left(x_{1,1},\ldots ,x_{1,L}\right),p_{2}=\left(x_{2,1},\ldots ,x_{2,L}\right)$$
with $x_{1,L}=x_{2,L}$Map each generated pair to a pair of latent variables using the inference model $q\left(z|x\right)∼N\left(\mu (x),\sigma (x)\right)$.
$$z_{1,1}=\mu \left(x_{1,1}\right),z_{2,1}=\mu \left(x_{2,1}\right)$$Compute the value of the absolute linear difference between the variables related to the sample:
$$e=(\left|z_{1,1}-z_{2,1}\right|,\ldots ,\left|z_{1,L}-z_{2,L}\right|)$$The mean of all pair differences in a batch gives a single instance in the final training set. These steps are repeated for each generative factor in order to create a substantial training set.Train a linear classifier on the generated training set to predict which generative factor has been fixed.$Z_{diff}$ score, also known as β-VAE metric, is the accuracy of the classifier.

In a perfectly disentangled representation, we would expect a zero in the dimension of the training input associated with the fixed generative factor, and the classifier would learn to map the zero-value index to the factor index.

### 4.2. $Z_{min}$ Variance Score

To overcome certain weaknesses of $Z_{diff}$ score, Kim and Mnih [42] introduced an unsupervised metric called *Z*-min Variance, also known as Factor-VAE metric. The intuition behind this metric is the same as the β-VAE metric with some improvements: a change in the way the latent representation is formed when a generative factor is fixed, in addition to the use of a specific type of classifier to predict which factor has been fixed. Calculating the Factor-VAE metric requires these steps:Randomly choose a generative factor $f_{k}$.Generate a batch of vectors, where the value of the selected factor $f_{k}$ is held fixed in the batch while the other generative factors $f_{k-1}$ are randomly selected. For a batch of *L* samples:
$$p_{1}=\left(x_{1,1},\ldots ,x_{1,L}\right)$$Map each generated vector to latent code using the inference model:
$$q\left(z|x\right)∼N\left(\mu (x),\sigma (x)\right)$$Normalize each variable within the latent representation using its empirical standard deviation calculated on the dataset. For a batch of *L* samples:
$$\left(z_{1}/s\ldots z_{L}/s\right)$$Calculate the empirical variance in each code of the normalized representations.
$$e=\left(Varz_{1}/s\ldots Varz_{L}/s\right)$$The factor index *k* and the latent variable index that has the lowest variance provide a training instance for the classifier. The factor index k and the index of the code dimension with the lowest variance give one training point for the classifier. These steps are repeated for each generative factor in order to create a substantial training set.Train a majority vote classifier on the generated training set to predict which generative factor was fixed.$Z_{min}$ Variance score is equivalent to the classifier accuracy.

By normalizing the latent representations, the authors ensure that the argmin is insensitive to the rescaling of the representation in each latent variable. For a perfectly disentangled representation, one expects to have an empirical variance of zero in the dimension corresponding to the fixed factor.

### 4.3. Mutual Information Gap (MIG Score)

Chen et al. [41] introduced a new disentanglement metric based on mutual information theory. Mutual Information Gap (MIG) computes the mutual information (MI) between each generative factor $x_{i}$ and latent code $z_{j}$. Higher mutual information denotes a deterministic relationship between $z_{j}$ and $x_{j}$. The mutual information gap score can be estimated through the steps below:Calculate the mutual information between each pair of latent variables and known generative factors.Each generative factor may have high mutual information with several latent variables. Therefore, for every single factor, classify latent variables according to the amount of information they stored about this factor.Calculate the difference between the top two values of mutual information for each generative factor.Normalize this difference by dividing by the entropy of the corresponding generative factor.The Mutual Information Gap (MIG) score is equivalent to the average of these differences [41]:
(26)$$MIG\left(x,z\right)=\frac{1}{K}\sum_{k}\frac{1}{H(x_{k})}\left(I\left(z_{j^{\left(k\right)}};x_{k}\right)-\max_{j\neq j^{\left(k\right)}}I\left(z_{j};x_{k}\right)\right)$$
where $j^{\left(k\right)}=\arg\max_{j}I\left(z_{j},x_{k}\right)$, and K is the known generative factors.

### 4.4. Attribute Predictability Score (SAP)

In parallel to the MIG score, Kumar et al. [43] provided a metric of disentanglement also based on Mutual Information. Applying this metric requires these steps:For each generative factor, compute the R2 score of linear regression (for continuous factors) or classification score (balanced accuracy for categorical factors) of predicting a *j*-th generative factor using only a *i*-th variable in the latent representation.Compute the difference between the top two most-predictive latent codes.The mean of those differences is the Attribute Predictability Score (SAP) [43].
(27)$$\begin{array}{c} SAP\left(x,z\right)=\frac{1}{K}\sum_{k}\left(I_{i_{K},K}-\max_{j\#i_{k}}I_{j,k}\right) \end{array}$$
where $i_{k}=\arg\max_{i}I_{i,k}$, and *K* is the number of known generative factors.

## 5. Discussion

In this section, we discuss the methods and metrics presented in this review and highlight associated opportunities and open challenges.

### 5.1. Methods

The computational methods aim to use variational encoding along with different ELBO decompositions to learn the disentangled representation of the data. The shared point between each of these methods is either up-weighting the VAE objective, Equation (Equation 10), or adding some regularizers to the VAE objective that act to match the approximate posterior $q_{\phi }\left(z\right)$ to the factorized prior $p_{\theta }\left(z\right)$. Figure 3 illustrates this idea:

By merely up-weighting the ELBO of VAE, the posterior $q_{\phi }\left(z|x\right)$ is pushed to correspond to the factorized prior $p_{\theta }\left(z\right)$, which results in a better disentanglement in comparison to the variational auto-encoder. β-VAE has shown acceptable performance. However, this penalization increases the tension between maximizing the data likelihood and disentanglement. As a result, there is a compromise between the accuracy of the reconstruction and the quality of disengagement within the latent representations learned. A higher value of β allows the achievement of better disentanglement but restricts latent channel information capacity and, therefore, a loss of information as it crosses this limited capacity latent *z*. The loss of high-frequency details about the data leads to poor reconstruction quality.

InfoMax-VAE outperforms β-VAE by constraining the latent representation so that the quantity of information kept about the observed data is maximized. In doing so, InfoMax-VAE is capable of obtaining high disentangling performance while maintaining a better reconstruction quality.

Factor-VAE ensures independence in the latent space by penalizing the total correlation term in the ELBO. A higher value of γ leads to a lower total correlation and, therefore, encourages independence in the code distribution. On the other hand, by not penalizing the mutual information, Factor-VAE keeps the information stored in z about x. Thus the model preserves high-frequency details about the data. By doing so, Factor-VAE improves upon β-VAE and has been reported to achieve a better balance between disentanglement and reconstruction quality. However, this model mostly remains difficult to train since it calls for an auxiliary discriminator and an internal optimization loop. On the other hand, the addition of a hyper-parameter during training may affect the model stability.

In β-TCVAE, the authors confirm the importance of total correlation for learning disentangled representation, and they propose an improvement in β-VAE and Factor-VAE. After breaking down the ELBO to a KL divergence term, mutual information and total correlation, they claim that adjusting β produces the best results. Thus they set α=γ=1, arriving at the same objective as Factor-VAE but with a simple way to estimate the total correlation without any additional hyper-parameter for more stable training. β-TCVAE has been capable of capturing independent factors in data distribution without having any degradation in the quality of reconstruction, thus surpassing β-VAE. However, we have the same disentanglement performance as Factor-VAE but in a simple manner to compute the total correlation.

DIP-VAE attempts to improve the performance of disentanglement by encouraging independence during inference. Having a disentangled prior that can be the basis for a generative disentangled model is the key idea behind DIP-VAE. DIP-VAE pushes the aggregated posterior $q_{\phi }\left(z\right)$ to correspond to a factorized prior p(z) by matching the moments of the two distributions. By doing so, DIP-VAE is capable of learning disentangled representation without introducing any trade-off between disentangling latent variables and maximizing the data likelihood. As a result, DIP-VAE has a better reconstruction quality contrary to β-VAE and with similar performance to Factor-VAE and β-TCVAE with learning disentangled representation without introducing a compromise between disentangling latent variables and the plausibility of the observed data.

In Table 4, we summarize the different choices of the regularizers applied for each method.

### 5.2. Metrics

It is currently unclear what exactly makes a disentangled metric better than another, but before analyzing metrics, we propose three criteria that constitute a disentangled representation, and we seek to analyze to what extent the metrics respect these criteria.

#### 5.2.1. Properties of a Disentangled Representation

Modularity: changes in one factor have no impact on other factors [90]. The same analogy is in the representation space, and the factors are also independent. This property is also known as disentanglement in [87].

Compactness: the extent to which each generative factor is entered by one latent variable [88]. In other words, varying an underlying factor should have as small as possible effect on the latent space. Ideally, each generative factor is associated with only one latent code. In [87], the author refers to these criteria as completeness.

Informativeness: the amount of information shared between latent variables and generative factors [87]. In other words, the value of a given factor can be precisely determined from the code. This criterion is also known as explicitness in [90].

#### 5.2.2. Comparison

Previous attempts to quantify disentangling have considered different aspects of modularity, compactness and informativeness criteria.

Higgins [20] argues that a good representation is one where the modularity of the latent representation holds. To make sure this property holds, he assumes that generative factors are independent and quantify the independence of the inferred latent variables using a simple classifier. However, this metric has several weaknesses. The $Z_{diff}$ score is based on the classifier used to achieve the score. Nevertheless, the classifier could be sensitive to several hyper-parameters, such as the choice of the optimizer, the initialization of the weights and the epochs. Furthermore, there may be a perfectly disentangled representation where a generative factor corresponds to several latent variables instead of one variable. Thus $Z_{diff}$ does not satisfy the compactness property. For example, if we fix this generative factor, the corresponding latent codes will have a variation of 0. Therefore, the classifier fails to distinguish between the latent codes and returns an accuracy of less than 1. Finally, the β-VAE metric does not require any assumptions about the factor-code relationship and therefore does not satisfy informativeness criteria.

$Z_{min}$ Variance addresses several issues of the β-VAE metric. For instance, a majority vote classifier that predicts the fixed generative factor according to the variation of the latent variables actually allows having fewer additional hyper-parameters to optimize and, therefore, having a more reliable final score. On the other hand, and similar to the β-VAE metric, $Z_{min}$ Variance depends on data with an independent factor. Furthermore, Zmin variance satisfies the compactness property by fixing the number of subsets of data and generating a training set that covers all possible combinations of factor-latent codes while maintaining a fixed factor. However, it does not require any assumptions about the factor-code relations. Thus it does not fulfill the informativeness criteria.

Unlike the $Z_{diff}$ or $Z_{min}$ metrics, the SAP score does not require any additional classifier and, therefore, returns a more reliable final score. The SAP metric satisfies informativeness and compactness criteria by computing the score of predicting each generative factor using a single variable in the latent representation. However, it does not penalize the modularity between generative factors.

The main advantage of the Mutual Information Gap (MIG) score is that it does not require many additional hyper-parameters contrary to β-VAE and Factor-VAE. Moreover, it encourages the compactness of the representation by pushing only one latent variable to be informative about a factor. By definition, it computes the amount of information shared between latent variables and generative factors.

Table 5 summarizes the findings from our analysis.

## 6. Conclusions and Future Directions

In this work, we conduct an extensive survey of disentangled representation learning through five state-of-the-art approaches focused on variational auto-encoders, alongside a detailed study on how to quantify disentanglement as well as conducting a comparison on the state-of-the-art of supervised disentanglement metrics.

We highlighted the underlying processes of disentangled representation learning methods driven by the development and deployment of computer vision algorithms using deep neural network approaches. In fact, each method considered herein is a variant of variational auto-encoder, and more precisely, they only vary the VAE objective, known as the Evidence Lower Bound Objective (ELBO). In particular, these methods either (i) up-weight the evidence lower bound, (ii) add a regularization to the ELBO that acts to match the approximated posterior to the factorized prior or (iii) combine a regularization and overweight the ELBO. By just up-weighting the evidence lower bound objective, one can observe a clear trade-off between disentanglement and reconstruction quality, which is the case for β-VAE. On the other hand, adding a regularizer to the evidence lower bound objective (Factor-VAE, InfoMax-VAE and DIP-VAE) or up-weighting ELBO while adding a regularizer (β-TCVAE) allows better disentanglement while preserving reconstruction quality.

Further, this study performs a comprehensive analysis and a fair comparison of the most well-known supervised disentanglement metrics. It considered three criteria that make one disentangled representation better than another (i) modularity, (ii) compactness and (iii) informativeness, and then compared metrics with respect to each criterion. We found that it is difficult to satisfy all three criteria at the same time. Most of the existing metrics meet one or two out of the three criteria. Only the mutual information gap score is robust to these criteria and able to give a general measure of the disentanglement quality.

However, some limitations remain unresolved and could be addressed in further work in order to improve the relevance of disentangled representation learning approaches in future work. (i) Despite the empirical success, existing disentangled representation learning approaches tend to ignore the latent variables and produce unrealistic, blurry samples with a significant reconstruction error when applied to complex datasets. There are several papers that discuss the issue of latent variable collapse [91,92,93,94], but more analysis on this issue is needed. (ii) Most of the approaches are based on VAE or GAN, more research on other potential models, e.g., diffusion model [95], would allow new ways for disentangled representation learning. (iii) Although disentangled representation learning has achieved several successes in generalization to unseen scenarios, state-of-the-art approaches for learning such representations have so far only been evaluated on small synthetic datasets. It will be interesting to explore the ability to generalize on complex real-world datasets. (iv) Finally, recent works discussed how to handle visual attacks and anomaly detection using dimensionality reduction, such as the SVD algorithm for neural networks and GAN [96,97,98]. Inspired by these works, it will be interesting to explore the application of disentangled representation learning while preventing visual attacks.

Moving forward, this survey is not only useful to provide insights for researchers that are currently working in the related area but can also be used as a basis for the implementation of new approaches. Our current goal is that we can build on these methods to develop an approach capable of identifying the underlying generative factors on more challenging datasets for a real-world application in an industrial environment.

## Figures and Tables

**Figure 1 sensors-23-02362-f001:**
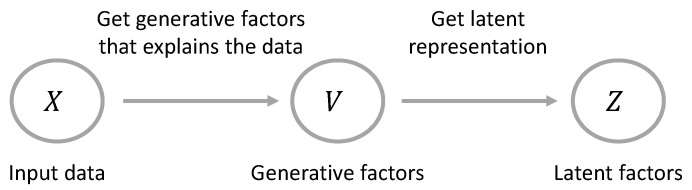
An illustration of the notation used in this paper. For $X=\left\{x_{1},x_{2},\ldots ,x_{N}\right\}$, a set of *N* observations. Disentangled representation learning is expected to identify the distinct generative factors $V=\left\{\nu_{1},\nu_{2}\ldots ,\nu_{n}\right\}$ that explain these observations and encode them with independent latent variables $Z=\left\{z_{1},\ldots ,z_{n}\right\}$ in latent space.

**Figure 2 sensors-23-02362-f002:**
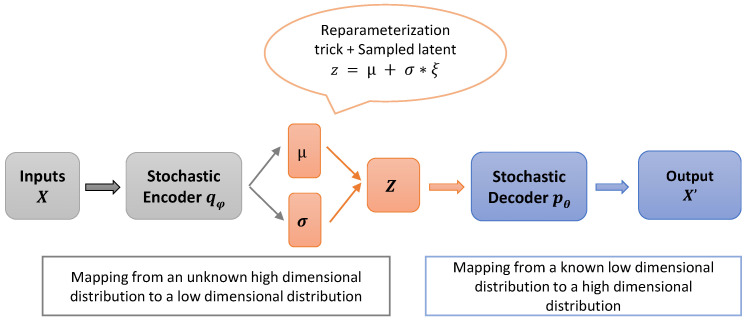
The structure of the variational auto-encoder (VAE). The stochastic encoder $q_{\phi }\left(z|x_{i}\right)$, also called the inference model, learns stochastic mappings between an observed *X*-space (input data) and a latent *Z*-space (hidden representation). The generative model $p_{\theta }\left(z|x_{i}\right)$, a stochastic decoder, reconstructs the data given the hidden representation.

**Figure 3 sensors-23-02362-f003:**
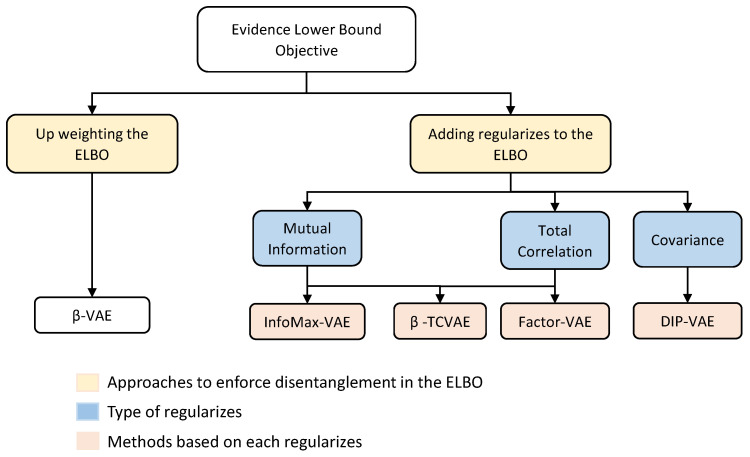
Schematic overview of different choices of augmenting the evidence lower bound of VAE. To improve disentanglement, most approaches focus on regularizing the original VAE objective by (i) up-weighting the ELBO with an adjustable hyperparameter β, resulting in a β-VAE approach. (ii) Adding different terms to the ELBO, such as mutual information, total correlation or covariance, resulting in InfoMax-VAE, β-TCVAE, Factor-VAE and DIP-VAE approaches, respectively.

**Table 1 sensors-23-02362-t001:** Overview of disentanglement methods based on auto-encoding and the metric reported with each method.

Methods	Metrics
β-VAE [20]	Z-diff Score
Factor-VAE [42]	Z-min Variance Score
β-TCVAE [41]	Mutual Information Gap (MIG)
DIP-VAE [43]	Attribute Predictability Score (SAP)
InfoMax-VAE [44]	-

**Table 2 sensors-23-02362-t002:** Overview of the main terms and corresponding mathematical expression.

Term	Mathematical Expression
Prior	$p_{\theta }\left(z\right)$
Generative Model (Decoder)	$p_{\theta }\left(z|x_{i}\right)$
Inference Model (Encoder)	$q_{\phi }\left(z|x_{i}\right)$
Data Log-Likelihood	$\log p_{\theta }\left(x_{i}\right)$
Kullback–Leibler Divergence	$D_{\mathrm{KL}}\left(q_{\phi }\left(z|x_{i}\right)\left|\right|p_{\theta }\left(z\right)\right)$
Evidence Lower Bound (ELBO)	$E_{q_{\phi }\left(z|xi\right)}\left[\log p_{\theta }\left(x_{i}|z\right)\right]-D_{\mathrm{KL}}\left(q_{\phi }\left(z|x_{i}\right)\left|\right|p_{\theta }\left(z\right)\right)$

**Table 3 sensors-23-02362-t003:** Details on the encoder and decoder architecture used to implement methods discussed in this paper depending on the chosen dataset.

Datasets	Encoder	Decoder
2D Shape	Input: 64 × 64 × number of channels,	Input: $R^{N}$, FC, 256 ReLU, FC,
3D Shape	Conv 32 × 4 × 4 (stride 2), 32 × 4 × 4 (stride 2),	4 × 4 × 64 ReLU, Upconv 64 × 4 × 4 (stride 2),
3D Chairs	64 × 4 × 4 (stride 2), 64 × 4 × 4 (stride 2),	32 × 4 × 4 (stride 2), 32 × 4 × 4 (stride 2),
3D Faces	FC 256, ReLU activation.	4 × 4 × number of channels (stride 2), ReLU activation. **Bernoulli Decoder**
CelebA	Input: 64 × 64 × 3, Conv 32 × 4 × 4 (stride 2), 32 × 4 × 4 (stride 2), 64 × 4 × 4 (stride 2), 64 × 4 × 4 (stride 2), FC 256, ReLU activation	$R^{N}$, FC, 256 ReLU, FC, 4 × 4 × 64 ReLU, Upconv 64 × 4 × 4 (stride 2), 32 × 4 × 4 (stride 2), 32 × 4 × 4 (stride 2), 4 × 4 × number of channels (stride 2), ReLU activation. **Gaussian Decoder**

**Table 4 sensors-23-02362-t004:** Summary of different ELBO decompositions alongside the regularization applied. For each method, the learning objective is given by $L_{common}+$
*regularizer*.

Method	$L_{\mathrm{common}}$	Regularizers
VAE	$L_{\mathrm{VAE}}$	−
β-VAE	$L_{\beta \text{-}\mathrm{VAE}}$	−
InfoMax-VAE	$L_{\beta \text{-}\mathrm{VAE}}$	$\alpha I_{q_{\phi }}\left(x,z\right)$
Factor-VAE	$L_{\mathrm{VAE}}$	$\mathrm{TC}\left(q_{\phi }\left(z\right)\right)$
β-TCVAE	$L_{\beta \text{-}\mathrm{VAE}}$	$\alpha I_{q_{\phi }}\left(x,z\right)$ + $\mathrm{TC}\left(q_{\phi }\left(z\right)\right)$
DIP-VAE	$L_{\mathrm{VAE}}$	$\lambda {∥{\mathrm{cov}}_{q_{\phi }\left(z\right)}\left[z\right]-I∥}_{F}^{2}$

**Table 5 sensors-23-02362-t005:** Summary of our findings from metric comparison. We found that most of the existing metrics can either satisfy modularity, compactness or informativeness criteria, except for the Mutual Information Gap score that captures all the three properties of a representation.

Metric	Satisfy Modularity	Satisfy Compactness	Satisfy Informativeness
$Z_{diff}$	Yes	No	No
$Z_{min}$	Yes	Yes	No
SAP	No	Yes	Yes
**MIG**	**Yes**	**Yes**	**Yes**

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
