# Peer review of "Unsupervised Learning of Disentangled Representation via Auto-Encoding: A Survey"

_sensors, 2023, doi:10.3390/s23042362_

Round 1

Reviewer 1 Report

This paper focuses on unsupervised representation learning and disentanglement in auto-encoders. This is an important targeted survey paper.

Comments:

* Authors should add a section to discuss how this survey positions with respect to self-supervised learning and unsupervised methods other than auto-encoding such as generative methods.

* In the introduction, authors should briefly discuss how disentanglement leads to generalizability, as claimed in the abstract of this manuscript.

* At the beginning of the introduction, the disentanglement aspect should be explained better along with examples.

* It looks authors have not revised the manuscript well, there are so many writing errors. There are dummy/placeholder texts around, multiple capitalization issues, spelling mistakes, missing references, and other grammatical mistakes. All these need to be corrected.

* The number of papers covered are very less, authors should do more literature survey.

* The table and figure caption should be more informative, and complete.

* Quantitative evaluation and detailed analysis of different kinds of disentanglements with various metrics is missing. Authors should include this in the revised version.

* Details of previous papers is occupying unnecessary space. Authors should remove the redundant details of these previous papers (reference them), and include only the discussion or essence of those papers which is relevant for writing this survey.

Overall the survey is very lightweight and does not provide a thought-provoking, in-depth discussion of the subject. For this reason, the authors are asked to increase the depth and width of this survey by incorporating the above-mentioned points. Below are some surveys to have the idea on the comprehensibility aspect of a survey.

[1] Schiappa, M. C., Rawat, Y. S., & Shah, M. (2022). Self-supervised learning for videos: A survey. ACM Computing Surveys.

[2] Shen, Z., Liu, J., He, Y., Zhang, X., Xu, R., Yu, H., & Cui, P. (2021). Towards out-of-distribution generalization: A survey. arXiv preprint arXiv:2108.13624.

Reviewer 2 Report

A detailed review of autoencoder is given in the paper. Metrics are analyzed too. Types of regularization are described.

The paper is well organized and presents state of the art overview.

However, the paper should be carefully read.

For example, l 56 - section number is missed. L 48 stated that this paper is review; however, the paper type is article.

The reference to formulas 25, 26 should be added.

Some references are not well presented (ref 38, for example)

From my point of view, it is important to describe also the problems with reconstruction error.

Reviewer 3 Report

The article is devoted to such an unsupervised learning method as encoders.

There are a number of comments on the work:

1) In the contact information of the authors, you use FirstName.LastName, you must specify real emails.

2) Key words should be added: generative models, neural networks.

3) Abstract: the results obtained during the analytical review should be detailed: what are the advantages of which models.

4) Line 16: Quoting [1-25]. It is necessary to break into a more detailed description of groups of articles. For example, in works [1-6] it was suggested… Works [7-15] are devoted to… The problem… is considered in [16-20]. In [21-25], it was possible to improve the methods ...

5) The introduction should mention simple dimensionality reduction models: PCA (https://doi.org/10.3390/rs10060907), SVD (https://doi.org/10.3390/app12083917), as well as methods for generating images from model parameters ( 10.1109/SYNCHROINFO49631.2020.9166000), and then go to the class of neural network algorithms (https://doi.org/10.3390/app9224780, https://arxiv.org/abs/2206.04452).

6) Line 42: Probably the link should be to table 1.

7) Line 57: section ?? -> section 5.

8) Check English. For example, line 116: as is -> as it is

9) Line 163: Link to table 5, but only 2 tables were entered before!

10) Line 226: Typo ratio trick [52?]

11) Line 243: The expression after it does not fit, and the number is not visible.

12) Line 272: Kumar and al -> Kumar et al.

13) Lines 279-280: Table ??

14) Line 293: Link to table 5, but only 3 tables entered before.

15) In the introduction, you should add tasks in which such models are applied. For example, super resolution (https://doi.org/10.3390/app12126067), or entertaining multimodal text-to-image models that are popular today (https://doi.org/10.3390/arts11050083).

16) Authors should add more detailed architectures of different encoders as pictures and give a brief description of them.

17) Line 466: There should probably be a link to table 5.

18) In Conclusion, it is necessary to note the directions for further additions to the review: GAN models (https://arxiv.org/abs/1406.2661, https://doi.org/10.3390/s22228761) and diffusion models (https://doi.org/ 10.3390/rs14194834) and also note that issues such as mod collapse (https://doi.org/10.1007/978-3-030-86340-1_45) and visual attacks (https://doi.org) can be considered. /10.3390/app11115235).

Round 2

Reviewer 1 Report

1. The reviewer doesn't agree with the author's response 1.1. : GANs have been shown to work better than VAE's and self-supervised methods are way better state-of-the-art in representation learning. Please reposition your arguments. Provide recent supportive evidence for your claims.

2. Author response 1.2, is not supported with any evidence or reference. For this reason, the authors should provide sufficient quantitative/qualitative details.

3. Author response 1.7: The manuscript is very brief in terms of the quality content that is expected from a survey (even from a brief survey). Authors should there include experimental studies to support their claims. At present state, the survey is more opinionated than factual.

4. Author response 1.8, covering the extended length of already published work without many insights is not beneficial for the research community. Authors should therefore provide only significant insights and drop redundant information because it overshadows the contributions of the survey.

5. The reviewer reiterates the final and still missing point: "Overall the survey is very lightweight and does not provide a thought-provoking, in-depth discussion of the subject. For this reason, the authors are asked to increase the depth and width of this survey by incorporating the above-mentioned points. Below are some surveys to have an idea of the comprehensibility aspect of a survey."

[1] Schiappa, M. C., Rawat, Y. S., & Shah, M. (2022). Self-supervised learning for videos: A survey. ACM Computing Surveys.

[2] Shen, Z., Liu, J., He, Y., Zhang, X., Xu, R., Yu, H., & Cui, P. (2021). Towards out-of-distribution generalization: A survey. arXiv preprint arXiv:2108.13624.

Reviewer 2 Report

The paper is improved. Thank you

Reviewer 3 Report

In general, the comments on the article were eliminated by the authors, however, in the opinion of the reviewer, it is important to note that the SVD algorithm is also useful in models for combating visual attacks, which would be useful to note in the introduction (https://doi.org/10.3390/app12083917) , as well as the use of Encoder and GAN algorithms.
